# Clinical and Cytogenetic Characterization of Early and Late Relapses in Patients Allografted for Myeloid Neoplasms with a Myelodysplastic Component

**DOI:** 10.3390/cancers14246244

**Published:** 2022-12-18

**Authors:** Victoria Platte, Anika Bergmann, Barbara Hildebrandt, Dagmar Wieczorek, Esther Schuler, Ulrich Germing, Jennifer Kaivers, Rainer Haas, Guido Kobbe, Thomas Schroeder, Christina Rautenberg

**Affiliations:** 1Department of Hematology, Oncology and Clinical Immunology, University Hospital Duesseldorf, Heinrich Heine—University, 40225 Duesseldorf, Germany; 2Institute of Human Genetics, Medical Faculty, Heinrich Heine—University, 40225 Duesseldorf, Germany; 3Department of Hematology and Stem Cell Transplantation, West German Cancer Center Essen, University Hospital Essen, 45147 Essen, Germany

**Keywords:** AML, MDS, relapse, transplant, cytogenetics

## Abstract

**Simple Summary:**

Relapse as the most common reason for treatment failure after allogeneic hematopoietic stem-cell transplantation (allo-SCT) in myeloid neoplasms usually occurs during the first year post-transplant, but several patients experience late relapse. This retrospective study aimed to characterize early and late relapses regarding clinical and cytogenetic parameters. Analyzing 91 consecutive patients, we demonstrated an improved overall survival for late compared with early relapsed patients and carved out cytogenetics and disease risk stratification at diagnosis as well as pretransplant strategy as major determinants for the timepoint of relapse. The impact of cytogenetics was highlighted by comparative karyotype analyses demonstrating a higher frequency of clonal evolution in early relapses. Improving knowledge about factors predicting early relapse may enable a selection of patients who may benefit from strategies to prevent or delay relapse, and identifying patients being at risk for late relapse may trigger prolonged surveillance strategies, including bone marrow biopsies and measurable residual disease (MRD) assessment.

**Abstract:**

An improved understanding of relapse kinetics is required to optimize detection and treatment strategies for the post-transplant relapse of myeloid neoplasms. Therefore, we retrospectively analyzed data from 91 patients allografted for MDS (*n* = 54), AML-MRC (*n* = 29) and chronic myelomonocytic leukemia (CMML, *n* = 8), who relapsed after transplant. Patients with early (<12 months, *n* = 56) and late relapse (>12 months, *n* = 35) were compared regarding patient-, disease- and transplant-related factors, including karyotype analyses at diagnosis and relapse. After a median follow-up of 17.4 months after relapse, late relapses showed improved outcomes compared with early relapses (2-yr OS 67% vs. 32%, *p* = 0.0048). Comparing frequency of distinct patient-, disease- and transplant-related factors among early and late relapses, complex karyotype (*p* = 0.0004) and unfavorable disease risk at diagnosis (*p* = 0.0008) as well as clonal evolution at relapse (*p* = 0.03) were more common in early than in late relapses. Furthermore, patients receiving transplant without prior cytoreduction or in complete remission were more frequently present in the group of late relapses. These data suggest that cytogenetics rather than disease burden at diagnosis and transplant-related factors determine the timepoint of post-transplant relapse and that upfront transplantation may be favored in order to delay relapse.

## 1. Introduction

Allogeneic hematopoietic stem-cell transplantation (allo-SCT) represents the only curative treatment option for myelodysplastic syndrome (MDS) and acute myeloid leukemia with myelodysplasia-related changes (AML-MRC), but relapse occurs in up to 50% of patients and thereby remains the most common reason for treatment failure associated with a dismal prognosis [1,2]. About 80% of relapses occur within the first year after transplant, whereas the remaining patients relapse considerably later [3]. While several reports have demonstrated that the time point of relapse after transplant is associated with prognosis, it may also reflect differences in disease biology between early and late relapses [4,5]. Several parameters, such as the presence of high-risk molecular-genetic features at diagnosis or refractory disease prior to transplant, were shown to be associated with a higher risk for post-transplant relapse in general [6,7,8,9]. However, data regarding the characterization of early and late relapses have been limited so far [10,11]. Therefore, our objective was to identify differences regarding patient-, disease- and transplant-related factors among patients who suffered early and late relapses and to obtain information on predictive factors for the one or the other. Such information may allow a better understanding of relapse kinetics and characteristics and help to gain insights into the underlying pathophysiology, but it is also mandatory to optimize strategies for disease surveillance and treatment. For that reason, we analyzed 91 patients, who were allografted for MDS, AML-MRC or chronic myelomonocytic leukemia (CMML) and experienced post-transplant relapse, regarding patient-, disease- and transplant-related parameters. Furthermore, we performed cytogenetic analyses at diagnosis and relapse to discover karyotype changes and to analyze their impact on relapse kinetics.

## 2. Materials and Methods

### 2.1. Patients and Study Design

Patients above the age of 18 years who received an allo-SCT for AML-MRC, MDS or CMML between 2004 and 2020 and who experienced post-transplant relapse of the underlying disease were included in this retrospective analysis. Other myeloid disorders, such as de novo AML and atypical chronic myeloid leukemia (aCML), were not included in this retrospective analysis. For the purpose of this analysis, patients were divided into those with either early (≤12 months after allo-SCT) and late relapses (any timepoint of relapse >12 months after allo-SCT), according to a commonly chosen interval [2,12,13]. Both groups were compared regarding patient-, disease- and transplant-related factors. Furthermore, we compared cytogenetics prior to transplant and at relapse, aiming to detect clonal evolution. Detailed patient, disease and relapse characteristics are given in Table 1 and Appendix A. The cytogenetics of each patient is listed in Appendix A. Written informed consent for the Duesseldorf MDS registry and/or MDS biobank (MDS registry 2018-331-FmB and MDS biobank 3768) was obtained from all patients. Furthermore, all patients gave written informed consent for the scientific use of their data in context of the German Registry for Stem Cell Transplantation.

### 2.2. Cytogenetic and Fluorescence In Situ Hybridization (FISH) Analyses

As part of our clinical standard, bone marrow (BM) samples were obtained prior to allo-SCT and at the time of relapse, and conventional metaphase cytogenetic and FISH analyses from these samples were performed at the Department of Human Genetics according to standardized protocols. These results were accessible for our comparative karyotype analyses at the timepoints “prior to allo-SCT” and “at relapse”. Furthermore, we differentiated between those showing karyotype changes and those without changes. Karyotype changes were categorized as either clonal evolution (appearance of new clonal alterations additionally to those documented at diagnosis), appearance of a new clone in case of normal cytogenetics at diagnosis or loss of an initially abnormal clone.

### 2.3. Definitions

Disease status prior to transplant and conditioning intensity was defined as previously described [14,15,16,17]. Hematologic relapse was diagnosed in the case of either the recurrence of MDS-typical dysplasia, an increase of at least 5% BM blasts or the presence of blasts in peripheral blood (PB) and/or any extramedullary disease. In the absence of any criteria defining hematologic relapse, molecular relapse was defined as the recurrence of disease-specific molecular or cytogenetic alterations, a decrease in donor chimerism <95%, a mixed XY-FISH of 4% residual recipient cells or an increase in PB *WT1*-mRNA expression above the cutoff of 50 copies per 10^4^
*ABL* copies confirmed not only in one but also in a second analysis after an interval of 2 to 4 weeks. Regarding pretransplant strategies, we distinguished between patients who received a pretreatment consisting either of an AML-like induction with anthracycline/cytarabine or of at least one course of hypomethylating agents (HMA) and those who received upfront transplantation after FLAMSA-based sequential conditioning.

### 2.4. Statistical Analysis

The data lock point at the end of observation was 1 November 2020. For the exploration of continuous variables, median and range were given, and potential differences were investigated using Mann-Whitney test. For categorical variables, frequencies were displayed, and cross-tabulation and Fisher’s exact *t*-test were used to estimate differences. The time-to-event curves were calculated using the Kaplan–Meier method for overall survival (OS), defined as time from relapse until death or date of last follow-up. For univariate comparison, the log-rank test was used. Multivariate analysis was performed using a multiple Cox regression model with a step-wise backward procedure and included only those variables influencing outcomes in univariate analysis with a *p*-value of <0.1. A *p*-value of <0.05 was considered to be statistically significant for all analyses. GraphPad Prism^®^ 5.01 (GraphPad Software Inc., La Jolla, CA, USA) and SPSS for Windows (SPSS Inc., Chicago, IL, USA) were used for all statistical analysis.

## 3. Results

### 3.1. Patient, Disease and Transplant Characteristics

A total of 91 adult patients with different subtypes of MDS and AML-MRC were included in this retrospective analysis. According to the World Health Organization (WHO) 2016 classification [15], eight patients (9%) suffered from MDS with multilineage or single lineage dysplasia (MDS-MLD/-SLD), 42 patients (46%) from MDS with excess of blasts (MDS-EBI/-EBII), four patients (3%) from other MDS (MDS unclassifiable/MDS with ringsideroblasts), eight patients (9%) from a chronic myelomonocytic leukemia (CMML) and 29 patients (32%) from AML-MRC at the primary diagnosis. The disease risk of patients with MDS, AML-MRC and CMML was estimated using the revised International Prognostic Scoring System (IPSS-R), European Leukemia Net (ELN) 2017 classification and CMML-Specific Prognostic Scoring System (CPSS), respectively [14,16,18]. As shown in Table 1, very good/good IPSS-R, favorable ELN and low CPSS risk groups were combined as one favorable disease risk group (*n* = 35, 38%); intermediate IPSS-R, intermediate ELN and intermediate I/II CPSS formed one intermediate disease risk group (*n* = 20, 22%); and high/very high IPSS-R, adverse ELN and high CPSS were summarized as one unfavorable disease risk group (*n* = 36, 40%). A total of 37 patients (41%) had a normal karyotype, whereas 50 patients (55%) had cytogenetic alterations, including 29 patients (36%) with complex karyotype cytogenetics. The median time from diagnosis to transplantation was 5.6 months (range 1.2 to 177.1). Regarding the pretransplant strategy, 16 patients (18%) received Azacitidine, 31 patients (34%) received an intensive cytoreductive chemotherapy and 44 patients (48%) received an upfront transplantation without prior disease-specific therapy. A total of 64 patients (70%) received reduced intensity conditioning, whereas 27 patients (30%) underwent allo-SCT after standard-dose conditioning. The majority of patients (*n* = 75, 83%) received a graft from an unrelated donor. In 90 patients (99%), peripheral blood stem cells were used as sample sources, and only in one patient was bone marrow used. Further patient, disease and transplant characteristics are listed in Table 1 and Appendix A.

### 3.2. Relapse Characteristics and Outcomes

For all patients, median time from allo-SCT to relapse was 5.5 months (range 0.5 to 109.8 months) and 30%, 55%, 62%, 80% and 100% relapsed within 3, 6, 12, 24 and >24 months, respectively (Figure 1). The majority of 68 patients (75%) suffered from hematologic relapse, whereas 23 patients (25%) were detected at the stage of molecular relapse. A total of 56 patients (62%) experienced early and 35 patients (38%) experienced late relapse after allo-SCT. Extramedullary disease relapse occurred in one patient. With a median follow-up of 17.4 months, the median overall survival (OS) from relapse for the entire cohort was 14.1 months (range 0.8–129.3 months) corresponding to a 2-year (2-yr) OS of 44% [95% CI 32–56%] (Figure 2a) after relapse. Patients with late relapse showed a significantly improved OS compared with patients with early relapse (2-yr OS 66.5% vs. 31.7%, *p* = 0.0048). Accordingly, the hazard to die after a late relapse was significantly lower than that after an early relapse (HR 0.36 [95%CI 0.19–0.67], *p* = 0.0048) (Figure 2b).

### 3.3. Comparison of Early and Late Relapses after Allo-SCT

For a better understanding of relapse biology, we compared patient-, disease- and transplant-related factors between patients with an early and those with late relapse (Table 1). The frequencies of molecular and hematologic relapses were evenly distributed within the early and the late relapsed patients (*p* = 0.8). Furthermore, the frequencies of AML-MRC and different MDS subtypes at diagnosis did not differ between these groups. Regarding transplant-related factors, no differences in conditioning intensity, donor type or HLA-matching between the groups of patients with early and late relapse were found (Table 1).

Differences between the group of patients with early and that with late relapse were observed in the presence of complex karyotype (cKT) with a significantly higher frequency of cKT in the subgroup of patients relapsing early (*p* = 0.0004), while the frequency of deletion 17p and somatic TP53 mutations were equally distributed between both groups. Furthermore, patients with unfavorable disease risk at diagnosis were more frequently present within the group of early relapses (*p* = 0.0008). Regarding pretransplant strategies, those patients who were refractory to chemotherapy (CTX) or hypomethylating agents (HMA) were more frequently present in the subgroup of early relapses, whereas patients who received upfront transplantation or proceeded to transplant in remission after CTX were more commonly present in the subgroup of late relapses (*p* = 0.02). Consequently, the median time to relapse significantly differed between refractory patients and those who received upfront transplantation (refractory: median 3.2 months, range 0.9–40; upfront: median 10.6 months, range 0.5–109.8; *p* = 0.04), whereas there was no difference between the latter and those undergoing allo-SCT in CR after CTX (CR after CTX: median 12.5 months, range 1.5–62, *p* = 0.82). In a multivariate analysis, upfront transplantation or proceeding to transplant in complete remission remained the only parameter associated with a higher likelihood for late than for early relapse (Table 2). On post-transplant relapse treatment, there were no differences in frequency of higher-intensity (e.g., intensive salvage chemotherapy, second allo-SCT) and lower-intensity (e.g., hypomethylating agents +/− DLI) approaches among the groups with early and late relapses (early relapses, intensive approach—15/56 pts (27%), nonintensive approach—41/56 pts (73%); late relapses, intensive approach—10/35 pts (29%), nonintensive approach—25/35 pts (71%), *p* = ns). However, regarding the response to relapse treatment, we observed a significantly higher complete remission rate in those patients relapsing late compared with those who experienced early post-transplant relapse (late relapses, responses 22/35 pts (63%), vs. early relapses, responses 20/56 pts (36%); *p* = 0.0171; Appendix A).

### 3.4. Comparative Karyotype Analyses

For only those 68 patients with hematologic relapse, sequential cytogenetic data for further analyses were available in 64 of them, while four patients (all of them belonging to the group of early relapses) had to be excluded from comparative karyotype analyses because of missing data. Overall, 41 patients (64%) exposed KT changes at relapse compared with diagnosis, whereas in 23 patients (36%) cytogenetic analyses at relapse did not reveal any changes, with nine patients having a normal and 14 patients exposing an abnormal KT at primary diagnosis and at relapse. In those 41 patients with KT changes, these were evenly distributed with 11, 13 and 13 patients exposing a new clone, showing either a clonal evolution or a loss of an abnormal clone, respectively. The four remaining patients harbored two changes. Regarding these four patients, one patient showed a new clone and clonal evolution, two patients exposed a new clone and a loss of an abnormal, and one patient had clonal evolution combined with a loss of a clone. Regarding time until relapse, KT changes more frequently occurred in the subgroup of patients with early (*n* = 28/37, 76%) compared with those with late (*n* = 13/27, 48%) relapse (*p* = 0.03, Figure 3).

## 4. Discussion

First, our results show a progressive decline of relapse incidence over the post-transplant time period. Although the majority of relapses occurred early after transplant, there was still a relapse incidence of 20% among those patients who survived relapse-free beyond 24 months post-transplant, which is in line with other analyses regarding relapse kinetics [3,10,11].

In accordance with most previous studies, we observed a significantly improved OS in patients with late relapse compared with those with early relapse [4,5,10,19]. This difference could be observed, although the disease burden at diagnosis (reflected by WHO disease categories/stage) and at relapse (reflected by medullary blast count) as well as the frequency of molecular versus hematologic relapse did not differ between the group with early relapse and that with late relapse. The improved OS after diagnosis of relapse in patients with late compared with those with early relapse may be explained by the higher response rate to salvage therapy in the group of late relapses (Appendix A). The latter may be related to the significantly lower frequency of complex karyotype cytogenetics and patients with unfavorable disease risk at diagnosis within the group of patients with late relapse. Our findings are in line with results from Yeung et al. [10] and again support the idea that disease biology reflected by cytogenetics rather than disease burden is one of the major factors determining both the timing of relapse and the response to relapse treatment.

While other transplant-related factors, such as donor type and conditioning intensity, influenced the timepoint of relapse in the retrospective analysis by Yeung et al. [10], these factors did not seem to impact post-transplant relapse kinetics in our analysis. However, we observed that pretransplant strategy might influence the timepoint of relapse post-transplant given that patients undergoing allo-SCT either in remission or upfront without prior cytoreduction tended to experience late relapse, whereas those proceeding to transplant with refractory disease experienced early relapse more frequently. This suggests that in the refractory patients, the conditioning is not able to control disease until a substantial graft-versus-leukemia effect by the donor immune system is implemented. Of note, high-risk cytogenetic and molecular features, e.g., complex karyotype and TP53-mutations, were evenly distributed between patients proceeding to transplant upfront or in remission and those with chemorefractory disease. These observations and previous work from our group [8,21] underline that in addition to baseline cytogenetics, the physicians’ choice of pretreatment strategy represents another major parameter influencing the timepoint of and outcome after relapse.

Additionally, we detected that karyotype changes at relapse more frequently occurred in the subgroup of patients with early (*n* = 28/37, 76%) than those with late (*n* = 13/27, 48%) relapse (Figure 3). These observations might support the idea of a chemotherapy-driven pretransplant selection of highly resistant, aggressive clones that cannot be eradicated by conditioning but that can clonally evolve and finally drive early post-transplant relapse [8,20]. In line with our previous work, these observations could argue for a concept of upfront transplantation after sequential conditioning, at least in patients with MDS and elevated blast counts (>10%) or low-blast count AML-MRC, to potentially avoid clonal selection and at least delay post-transplant relapse to a later period of time after transplant, when relapsed patients may again better tolerate and respond to salvage therapy and donor cell–based consolidation with DLI or a second transplant [8,21]. We are aware that at this point, this idea remains hypothetical and requires confirmation in a prospective trial, ideally comparing the concept of upfront transplantation with new treatment options, such as CPX-351 or HMA/Venetoclax, which were shown to improve outcomes, especially in allografted patients [22,23,24,25,26]. In addition to the intrinsic limitations of a retrospective analysis on potential selection bias and the lack of longitudinal sampling during the course of disease and the follow-up period post-transplant, especially the lack of a comprehensive molecular dataset has to be taken into account as a major limitation when interpretating the results of our current analysis.

## 5. Conclusions

In conclusion, our data suggest that in MDS/AML-MRC, pretransplant strategy as well as genetics rather than disease phenotype/blast count at diagnosis or transplant-related factors determine the timing of post-transplant relapse. The impact of cytogenetics is also reflected by higher genomic instability, causing a higher frequency of clonal evolution in those with early relapse. Improving knowledge about factors predicting early relapse may enable a selection of patients who may benefit from preventive strategies, such as post-transplant maintenance therapy, prophylactic DLI or the early tapering of immunosuppression in order to prevent or at least delay relapse and improve patient survival. On the other hand, identifying patients who are at risk for late relapse may trigger a prolonged surveillance strategy, including bone marrow biopsies and MRD assessment.

## Figures and Tables

**Figure 1 cancers-14-06244-f001:**
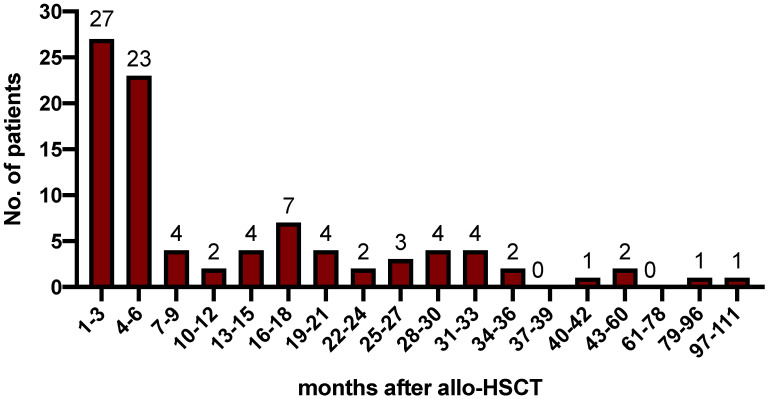
Timepoint of post-transplant relapses. Median time from allo-SCT to relapse was 5.5 months (range 0.5 to 109.8 months), and 30%, 55%, 62%, 80% and 100% relapsed within 3, 6, 12, 24 and >24 months, respectively. No., number.

**Figure 2 cancers-14-06244-f002:**
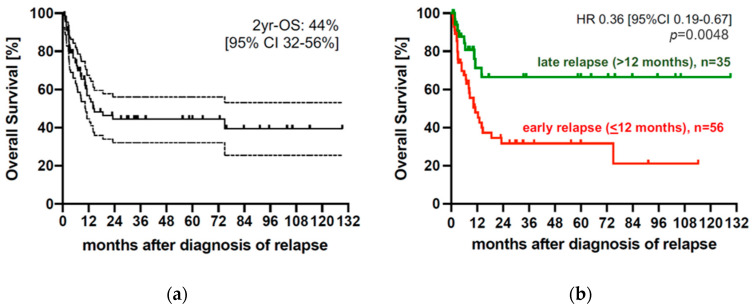
Outcome after post-transplant relapse. (**a**) Overall survival of the whole cohort (*n* = 91), median OS of the whole cohort was 14.1 months (range 0.8–129.3 months) corresponding to a 2-yr OS of 44% [95% CI 32–56%] after relapse. (**b**) Outcome of patients with early and late relapses, patients with late (green line, *n* = 35) relapse showed a significantly improved outcome compared with patients with early (red line, *n* = 56) relapse (2-yr OS 66.5% vs. 31.7%, *p* = 0.0048). Accordingly, the hazard to die after a late relapse was significantly lower than that after an early relapse (HR 0.36 [95%CI 0.19–0.67], *p* = 0.0048). Allo-SCT, allogeneic hematopietic stem-cell transplantation; HR, hazard ratio; CI, cumulative incidence; OS, overall survival; yr, year.

**Figure 3 cancers-14-06244-f003:**
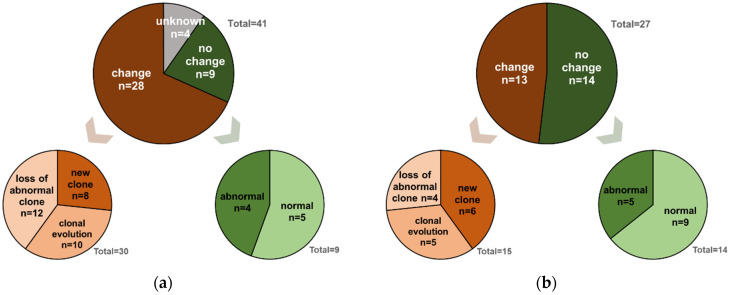
Comparative karyotype analyses prior to allo-SCT and at the time of early relapse (**a**) or late relapse (**b**): (**a**) relapse ≤12 months post allo-SCT, (**b**) relapse >12 months post allo-SCT. Karyotype changes more frequently occurred in the subgroup of patients with early (*n* = 28/37, 76%) than those with late (*n* = 13/27, 48%) relapse (*p* = 0.03). The proportion of patients showing a different karyotype in a relapse sample compared with a sample of prior-to allo-SCT was depicted in dark brown (=change), while the proportion of patients whose karyotype did not change at relapse compared with a pretransplant sample was illustrated in dark green (=no change). Karyotype changes were further described as either clonal evolution (appearance of new clonal alterations in addition to those documented at diagnosis), appearance of a new clone or a loss of an abnormal clone and were depicted in different shades of brown. In total, four patients developed two karyotype changes and therefore appeared twice in the descriptive subgroups. Within the early relapses, one patient harbored a combination of a new clone/clonal evolution and another patient a clonal evolution/loss of abnormal clone. Within the late relapses, two patients developed both a new clone/loss of abnormal clone. Patients without karyotype changes were further subdivided into those exposing a normal or an abnormal karyotype, which were illustrated in different shades of green; allo-SCT, allogeneic hematopoietic stem-cell transplantation.

**Table 1 cancers-14-06244-t001:** Patient and disease characteristics.

Characteristic	All	Early Relapse (≤12 Months)	Late Relapse (>12 Months)	*p*
No. of patients (%)	91	56 (62)	35 (38)	
**Age at transplant,** **median (range), year**	57 (20–76)	59 (20–71)	57 (24–76)	ns
**Gender**				
Male	22 (48)	37 (66)	20 (57)	ns
Female	24 (52)	19 (34)	15 (43)	
**WHO 2016 diagnosis** (15)				
AML-MRC	29 (32)	21 (38)	8 (23)	ns ^a^
MDS-SLD/-MLD	8 (9)	4 (7)	4 (11)	ns ^b^
MDS-RS-SLD/-MLD	1 (1)	0 (0)	1 (3)	
MDS-EBI/-II	42 (46)	22 (39)	20 (57)	ns ^c^
MDS-u	3 (3)	3 (5)	0 (0)	
CMML-0	1 (1)	1 (2)	0 (0)	ns ^d^
CMML-I/-II	7 (8)	5 (9)	2 (6)	
t-related myeloid neoplasm (MDS)	12 (13)	7 (13)	5 (14)	
**Karyotype at diagnosis**				
Normal	37 (41)	20 (36)	17 (49)	ns
Aberrant	50 (55)	32 (57)	18 (41)	
Complex	29 (36)	25 (45)	4 (11)	**0.0004** ^e^
Missing	4 (4)	4 (7)	0	
**ELN 2017//IPSS-R//CPSS** **at diagnosis** (14,16,18) *				
favorable	35 (38)	15 (27)	20 (57)	
intermediate	20 (22)	11 (20)	9 (26)	**0.0008** ^f^
unfavorable	36 (40)	30 (53)	6 (17)	
**HCT-CI** (19)				
Low	43 (47)	25 (45)	18 (51)	ns
Intermediate/high	48 (53)	31 (55)	17 (49)	
**Conditioning** (17)				
Standard-dose	27 (30)	16 (29)	11 (31)	ns
Dose-reduced	64 (70)	40 (71)	24 (69)	
**Donor Type (*n* = 90)**				
related	15 (17)	12 (22)	3 (9)	ns
unrelated	75 (83)	43 (78)	32 (91)	
**HLA-matching**				
matched	61 (67)	37 (66)	24 (69)	ns
mismatched	30 (33)	19 (34)	11 (31)	
**Type of Relapse**				
molecular	23 (25)	15 (27)	8 (23)	ns
hematolgic	68 (75)	41 (73)	27 (77)	
**Karyotype at relapse (*n* = 68 ^&^)**				
change	41 (64)	28 (76)	13 (48)	0.03
no change	23 36)	9 (24)	14 (52)	
**Blast count at relapse**				
normal	36 (40)	21 (38)	15 (43)	ns
elevated	55 (60)	35 (62)	20 (57)	
**Pretreatment Strategy**				
Upfront ^#^ or CR after CTX	51 (56)	26 (46)	25 (71)	**0.02**
no CR after CTX	40 (44)	30 (54)	10 (29)	

* very good/good IPSS-R, favorable ELN and low CPSS risk group were combined as one favorable disease risk group, while intermediate IPSS-R, intermediate ELN and intermediate I/II CPSS formed one intermediate disease risk group and high/very high IPSS-R, adverse ELN and high CPSS were summarized as one unfavorable disease risk group. ^#^ upfront: patients who proceeded to transplantation without prior cytoreductive chemotherapy defined as no MDS- or AML-MRC-specific therapy except for growth factors, transfusions and a short period (<1 week) of hydroxyurea. ^&^ only hematologic relapses were included. ^a^ AML-MRC vs. all others. ^b^ MDS-SLD/-MLD/-RS vs. all others. ^c^ MDS-EB1/-EB2 vs. all others except AML-MRC and MDS-EB1/-EB2 vs. all others, including AML-MRC. ^d^ CMML vs. all others. ^e^ cKT vs. all other karyotypes, including normal KT. ^f^ favorable/intermediate vs. unfavorable. The disease risk of patients with MDS, AML-MRC and CMML was estimated using IPSS-R, ELN 2017 classification and CPSS, respectively, and the respective subgroups were summarized as described above. AML-MRC, acute myeloid leukemia with myelodysplasia-related changes; CMML-0/-I/-II, chronic myelomonocytic leukemia; CPSS, CMML-specific scoring system; CR, complete remission; CTX, chemotherapy; ELN, European leukemia net; HCT-CI, hematopoietic cell transplant comorbidity index; HLA, human leukocyte antigen; int, intermediate; IPSS-R, International Prognostic Scoring System—revised version; MDS-EBI-/EBII, myelodysplastic syndrome with excess of blasts I/II; MDS-MLD, myelodysplastic syndrome with multilineage dysplasia; MDS-SLD, myelodysplastic syndrome with single lineage dysplasia; MDS-RS-MLD, myelodysplastic syndrome with multilineage dysplasia and ring sideroblasts; MDS-RS-SLD, myelodysplastic syndrome with single lineage dysplasia and ring sideroblasts; MDS-u, myelodysplastic syndrome unclassifiable; No., number; WHO, World Health Organization.

**Table 2 cancers-14-06244-t002:** Predictors for late relapse—multinominal logistic regression.

Variable	Odds Ratio ^”^	95% CI	*p*
**Treatment prior to allo-SCT** Upfront ^#^/CR vs. No CR	3.402	1.140–10.155	**0.028**
**ELN 2017//IPSS-R//CPSS** **at diagnosis** (14,16,18) * Favorable/intermediate Unfavorable	2.777	0.755–10.213	0.124
**Karyotype** Not complex vs. Complex	3.038	0.695–13.278	0.140

**^”^** Odds ratio represents odds of late relapse versus odds of early relapse among patients who experienced relapse. **^#^** upfront: patients who proceeded to transplantation without prior cytoreductive chemotherapy, defined as no MDS- or AML-MRC-specific therapy except for growth factors, transfusions and a short period (<1 week) of hydroxyurea. * very good/good IPSS-R, favorable ELN and low CPSS risk groups were combined as one favorable disease risk group; intermediate IPSS-R, intermediate ELN and intermediate I/II CPSS formed one intermediate disease risk group; and high/very high IPSS-R, adverse ELN and high CPSS were summarized as one unfavorable disease risk group. The disease risk of patients with MDS, AML-MRC and CMML was estimated using IPSS-R, ELN 2017 classification and CPSS, respectively, and the respective subgroups were summarized as described above. Allo-SCT, allogeneic hematopoietic stem-cell transplantation; CI, confidence interval; CPSS, CMML-specific scoring system; CR, complete Remission; int, intermediate; ELN, European Leukemia Net; IPSS-R, International Prognostic Scoring System—Revised Version.

## Data Availability

The data presented in this study are available on request from the corresponding author. The data are not publicly available, because the containing information could compromise the privacy of research participants.

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
