# Peer review of "Clinical and Cytogenetic Characterization of Early and Late Relapses in Patients Allografted for Myeloid Neoplasms with a Myelodysplastic Component"

_cancers, 2022, doi:10.3390/cancers14246244_

Round 1
Reviewer 1 Report
This is an interesting field. However, major issues need to addressed:
1. Introduction should include the objective or hypothesis of the study.
2. The number of patients is a result of a study and not a methodology. What were the inclusion/exclusion criteria? Which time period,?
3. Figure 2 should be self explanatory. What does change, no change mean?
4. The authors discuss about genomic instability, although no genetic testing is mentioned. Please clarify.
5. A paragraph of limitations is needed.
6. The reference list is limited. Additional references exist on real world of AMLMRC that are not cited.
Author Response
Thank you very much for these helpful suggestions.
Please see the attachment.
Kind regards, Christina Rautenberg

Reviewer 2 Report
The authors describe differences between patients with myeloid neoplasms who relapse early versus those who relapse late. The topic is particularly important since relapse is so common and carries with it such a poor prognosis.
There are some issues with the manuscript as it stands.
The paper is written rather well but a few sentences are constructed in a fashion that native English speakers will find somewhat awkward.
Many abbreviations are used throughout and it would be helpful to introduce all of them before using them, given that many readers will be international.
The definition of early versus late relapse could be made more clear by specifying the number of months instead of the number of years; < or = to 12 months is clearer than < or = 1 year. Is greater than 1 year, 2 years or more? Or greater than 12 months?
The title of the manuscript does not seem correct nor does it match the title of the supplemental material--the latter says "allografted for MDS and sAML." Do you mean secondary AML, like therapy-related? What about the CMML cases? CMML is a hybrid myelodysplastic and myeloproliferative neoplasm. Therefore, it is neither an MDS nor an AML, although it does have a myelodysplastic component. Would a more appropriate title include the phrase "myeloid neoplasms with a myelodysplastic component?"
From the abstract, it looks like there are 54 cases of MDS, 29 cases of AML, and 8 cases of CMML, for a total of 91 patients. However, in section 3.1, lines 107 to 112, it looks like there are a total of 54 MDS patients of various subtypes, 31 patients with AML, and 9 with CMML, for an overall 94 patients.
I am not that defined a molecular relapse as donor chimerism falling below 95% or having 4% or more sex-mismatch on FISH should constitute a molecular relapse. It reflects mixed chimerism for sure, however, it seems a little bit of a stretch to conclude that after a stem cell transplant, the donor cells must be neoplastic.
In table 1, the p-value for karyotype at relapse is 0.05 but it is in bold type--but the authors define statistical significance as less than 0.05. 0.05 could be 0.054 or it could be 0.049. Therefore, this needs to be clarified.
In table 2, the p-value for pre-treatment upfront/CR versus no CR is 0.028. Shouldn't this be in bold type? It looks significant and therefore should be highlighted by the authors to maintain consistency with the other table.
Overall, this is an important paper but it could be stronger if these issues were addressed.
Author Response

(The authors gave the same response as above.)

Reviewer 3 Report
Platte et al retrospectively analyze a cohort of 91 patients that underwent a stem cell transplant at their hospital for relapse criteria. They established that genetic factors way in more heavily than disease phenotype and blast percentage to determine SCT timeline.
This is a well written manuscript. It is well thought out and planned. Literature is well cited.
I have a few minor comments:
1. Please make the simple summary more readable. There are many long sentences with commas, obliterating the very reason for it.
2. Did only 91 patients receive allo-SCT since 2004? Were there more patients? If so, why were they excluded?
3. Figure 3 bottom has very light colours and the text is white, I recommend changing colour text to black for contrast.
Author Response

(The authors gave the same response as above.)

Round 2
Reviewer 3 Report
The authors have revised the manuscript according to reviewer comments.